# Coronary Artery Disease is More Severe in Patients with Non-Alcoholic Steatohepatitis than Fatty Liver

**DOI:** 10.3390/diagnostics10030129

**Published:** 2020-02-26

**Authors:** Toshihiro Niikura, Kento Imajo, Anna Ozaki, Takashi Kobayashi, Michihiro Iwaki, Yasushi Honda, Takaomi Kessoku, Yuji Ogawa, Masato Yoneda, Hiroyuki Kirikoshi, Satoru Saito, Atsushi Nakajima

**Affiliations:** 1Department of Gastroenterology and Hepatology, Yokohama City University Graduate School of Medicine, 3-9 Fukuura, Kanazawa-ku, Yokohama 236-0004, Japan; colorheartdarkness@gmail.com (T.N.); kento318@yokohama-cu.ac.jp (K.I.); anx0513ro@hotmail.com (A.O.); tkhkcb@gmail.com (T.K.); michihirokeidai@yahoo.co.jp (M.I.); y-honda@umin.ac.jp (Y.H.); takaomi0027@gmail.com (T.K.); yuji.ogawa01@gmail.com (Y.O.); dryoneda@yahoo.co.jp (M.Y.); ssai1423@yokohama-cu.ac.jp (S.S.); 2Department of Gastroenterology and Hepatology, Omori Red Cross Hospital, 4-30-1, Chuo, Ota-ku 194-0023, Japan; 3Department of Clinical Laboratory, Yokohama City University Hospital, 3-9 Fukuura, Kanazawa-ku, Yokohama 236-0004, Japan; hkirikos@med.yokohama-cu.ac.jp

**Keywords:** non-alcoholic fatty liver, non-alcoholic steatohepatitis, coronary CT angiography, coronary artery lesion, Agatston score

## Abstract

Non-alcoholic fatty liver disease (NAFLD) is associated with a higher risk of atherosclerotic disease. However, the relationships between the severity of coronary atherosclerosis and pathologic findings in patients with NAFLD remain unknown. We aimed to characterize the coronary artery lesions in patients with NAFLD using coronary computed tomography angiography (CCTA). Overall, 101 patients with liver biopsy-proven NAFLD who had chest pain or electrocardiographic abnormalities underwent CCTA. Coronary artery lesions, including coronary artery stenosis (CAS), calcium score (CACS, Agatston score), and coronary artery non-calcified plaque were assessed using multi-slice CT. Multivariate analysis showed that age, smoking status, prevalence of dyslipidemia (DLP) and non-alcoholic steatohepatitis (NASH), and stage of fibrosis were independent risk factors for CAS. Age, and the prevalence of DM and DLP, were independent risk factors for CACS, and the prevalence of NASH tended to be an independent risk factor. In addition, the prevalence of DLP and NASH were independent risk factors for non-calcified plaques. Coronary artery lesions are more common in patients with NASH than in those with non-alcoholic fatty liver, suggesting a higher risk in patients with NASH. Therefore, patients with NASH should be closely followed, with particular vigilance for coronary artery diseases.

## 1. Introduction

Non-alcoholic fatty liver disease (NAFLD) is generally considered to be the hepatic manifestation of metabolic syndrome and is characterized by fat accumulation in the liver in > 5% of hepatocytes, which occurs in the absence of significant alcohol intake, viral infection, or any other specific etiology of liver disease. NAFLD comprises a range of conditions from non-alcoholic fatty liver (NAFL) to non-alcoholic steatohepatitis (NASH) and cirrhosis. NAFLD is associated with a higher prevalence of atherosclerotic diseases, including cardiovascular disease (CVD) and diabetes mellitus (DM) [1,2]. Moreover, patients with NAFLD have higher risks of carotid intima media thickening [3,4], impaired endothelial function [5], coronary artery calcification [6,7], and greater arterial stiffness [8]. However, there are no data regarding the associations between the pathologic features of NAFLD and coronary artery lesions.

Coronary CT angiography (CCTA) has recently been established as a useful technique for evaluating coronary atherosclerosis, including stenosis, plaque characteristics, and calcium score (Agatston calcium score) [9]. Patients with high-risk coronary plaques, including non-calcified plaques, have a high risk of cardiovascular events, and the calcium score is proportional to the risk of such cardiovascular events [10,11]. NAFLD is associated with the progression of coronary calcium scores and high-risk coronary plaques, independent of conventional cardiovascular risk factors [12]. To date, however, no studies have investigated whether the severity of coronary artery lesions differs between patients with NAFL or NASH. We previously reported that the low-density lipoprotein (LDL)-migration index (MI), an indicator of serum small dense LDL (sdLDL) concentration, is higher in patients with NASH than in those with NAFL, which corresponds to the alterations in lipid metabolism involved, and this suggests that the risk of atherosclerotic diseases may be higher in patients with NASH than in those with NAFL [13,14]. Therefore, we hypothesized that there is a higher risk of coronary artery disease (CAD) in patients with NASH than in those with NAFL. This study aimed to assess coronary artery stenosis (CAS), coronary artery calcium score (CACS, Agatston score), and non-calcified plaques using CCTA in patients with NAFL or NASH.

## 2. Patients and Methods 

### 2.1. Ethical Approval of the Study Protocol

The protocol of this study conforms to the ethical guidelines of the 1975 Declaration of Helsinki as reflected in a priori approval by the institution’s human research committee, and the study was conducted with the approval of the Ethics Committee of Yokohama City University Hospital (Yokohama, Japan) on February 15, 2014. This trial is registered with the University Medical Information Network Clinical Trials Registry as UMIN000009614. All participants provided written informed consent before examination.

### 2.2. Study Population

The study population comprised patients who underwent liver biopsy between March 2014 and September 2018 and who were followed up at Yokohama City University Hospital in Japan. This study enrolled 114 patients, of whom five were excluded because they underwent percutaneous coronary intervention before CCTA examinations. We also excluded eight patients because of errors in CCTA examinations (six patients) or withdrawal of consent (two patients). Finally, 101 patients were enrolled in this study. The time interval between CCTA and liver biopsy was < 6 months. The exclusion criteria were as follows: any history of significant alcohol intake (≥210 g/week in men and 140 g/week in women), history of any other specific etiology of liver disease, or history of angina or myocardial infarction. After excluding 13 participants, data of the remaining 101 patients were analyzed (Figure 1).

### 2.3. Clinical and Laboratory Measurements

Basic demographic data, including the age and sex of study participants, and relevant medical history, including angina, myocardial infarction, renal dysfunction, DM, hypertension (HT), or dyslipidemia (DLP), were acquired from medical records. Smoking status and alcohol intake were obtained from the responses given in a self-reported questionnaire. The smoking score was calculated as the number of cigarettes smoked per day × the number of years as a smoker. Height and body mass were measured with participants wearing light clothing and no shoes. Body mass index (BMI) was calculated as body mass in kilograms divided by the square of height in meters (kg/m2). Blood pressure was measured after the participants rested quietly using an automatic manometer and an appropriate cuff at the time of hospitalization. Blood sampling was performed early in the morning, and the samples were analyzed in the central laboratory. Aspartate aminotransferase, alanine aminotransferase, γ-glutamyl transpeptidase, C-reactive protein, creatinine, fasting blood glucose, fasting insulin, hemoglobin A1c, fasting total cholesterol, triglycerides, high-density lipoprotein-cholesterol, and low-density lipoprotein-cholesterol concentrations were measured. DM was defined as a fasting plasma glucose concentration ≥126 mg/dL, self-reported history of DM, treatment with dietary modification, or use of antidiabetic medication. HT was defined as a blood pressure ≥ 140/90 mmHg or self-reported history of HT and/or the use of antihypertensive medication. DLP was defined as a total cholesterol concentration ≥ 240 mg/dL or the use of antihyperlipidemic medication.

### 2.4. Histopathologic and Immunohistochemical Evaluations

Liver biopsy specimens were obtained using a 16-gauge needle biopsy kit (Pro-Mag™ Ultra Automatic Biopsy Instrument; Argon Medical Devices, Frisco, TX, USA). Two specimens were obtained from each patient to acquire a sufficient sample size for analysis and to reduce histologic analysis errors. An adequate liver biopsy sample was defined as being > 20 mm in length and/or containing >10 portal tracts. Each specimen was assessed histologically by one pathologist, and the NAFLD activity score (NAS) was calculated [15]. Patients with steatosis, inflammation, and ballooned hepatocytes were classified as having NASH based on the results of the Fatty Liver Inhibition of Progression (FLIP) algorithm [16]. The severity of fibrosis was scored as previously described [17].

### 2.5. CCTA

CT images were acquired using a single-source 64-detector multi-slice CT with a standard scanning protocol. The protocol comprises a non-enhanced calcium score scan, followed by a contrast-enhanced CCTA scan. The images were then analyzed using a dedicated workstation (Volume analyzer, Synapse Vincent, Fujifilm, city, country). Based on the guidelines of the American Heart Association, a 16-segment coronary artery tree model was used. Using the Agatston score, coronary artery wall calcification was reported as previously described [18]. Agatston score is a semi-automated tool to calculate a score based on the extent of coronary artery calcification detected by an unenhanced CT scan, which is routinely performed in patients undergoing CCTA. This score was calculated using a weighted value assigned to the highest density of calcification in a given coronary artery and was stated in Hounsfield units (HUs). This weighted score was then multiplied by the area (in square millimeters) of the coronary calcification. The recently proposed Society of Cardiovascular Computed Tomography grading scale for CAS severity was used to assess the degree of luminal stenosis [19], in which Grade 0 represented no visible stenosis, Grade 1 represented 1–24% (minimal stenosis), Grade 2 represented 25–49% (mild stenosis), Grade 3 represented 50–69% (moderate stenosis), Grade 4 represented 70–99% (severe stenosis), and Grade 5 represented 100% (occlusion). The features of high-risk (non-calcified) plaques were defined as positive remodeling, CT attenuation < 30 HU, “napkin-ring” sign, and only spots of calcium deposition.

### 2.6. Statistical Analysis

Continuous variables are summarized as means and standard deviations and categorical variables as frequencies and percentages. Data were expressed as mean ± S.D., unless indicated otherwise. All statistical analyses were performed using SPSS 12.0 software (SPSS Inc., Chicago, IL, USA). The t-test and analysis of variance with Scheffe’s multiple testing correction were used for univariate comparisons between groups. Multivariate analysis was performed to determine factors independently associated with CAS, CACS, or non-calcified plaques. *p* <0.05 was considered to represent statistical significance.

## 3. Results

### 3.1. Baseline Characteristics

The study population comprised 101 patients with NAFLD (41 patients with NAFL and 60 with NASH). Baseline characteristics of each group are shown in Table 1. 

The mean age of the NAFL group (52.7 ± 13.9 years) was significantly lower than that of the NASH group (59.6 ± 11.2 years) (*p* = 0.009). In addition, there were significantly more women in the NASH group (men/women, 26/34) than in the NAFL group (24/17) (*p* = 0.0471). No differences in other parameters were identified between the NAFL and NASH groups. Histologic characteristics are also summarized in Table 1. The grades for lobular inflammation, hepatocyte ballooning, and fibrosis stage in the NASH group were higher than those in the NAFL group.

### 3.2. Associations between Coronary Artery Lesions and Histologic Findings

The associations of CAS, CACS, and non-calcified plaques with histologic characteristics are shown in Figure 2, Figure 3 and Figure 4, respectively. The severity of CAS, CACS, and non-calcified plaques was positively correlated with the hepatocyte ballooning grade and the stage of liver fibrosis (Figure 2, Figure 3 and Figure 4). However, there were no associations between these findings and the grades of steatosis or liver inflammation (Figure 2, Figure 3 and Figure 4).

### 3.3. Multiple Regression Analysis of Potential Risk Factors for CAS, CACS, and non-Calcified Plaque

The characteristics of NAFLD patients with and without CAS are shown in Table 2. Fifty-one patients had no significant stenosis and 50 had CAS. Univariate analysis showed that age; smoking score; prevalence of DM, HT, DLP, and NASH; and fibrosis stage were higher (*p* < 0.05 each) in patients with CAS than in those without. Multivariate analysis showed that age, smoking score, prevalence of DLP and NASH, and fibrosis stage were independent risk factors for CAS (all *p* < 0.05).

Next, the characteristics of NAFLD patients without (score = 0 AU) or with CACS (score >0 AU) were compared (Table 3). In total, 50 patients had no CACS and 51 had CACS. Univariate analysis showed that age; prevalence of DM, HT, DLP, and NASH; and fibrosis stage were higher (all *p* < 0.05) in patients with CACS than in those without. Multivariate analysis showed that age and the prevalence of DM and DLP were independent risk factors for CACS (*p* < 0.05 each). NASH tended to be associated with the presence of CACS, although not significantly (*p* = 0.0987).

Finally, the characteristics of NAFLD patients with and without non-calcified plaque are shown in Table 4. Sixty-three patients had no non-calcified plaque and 38 had non-calcified plaque. Univariate analysis showed that age; smoking score; prevalence of DM, HT, DLP, and NASH; and fibrosis stage were higher (*p* < 0.05 each) in patients with non-calcified plaque than in those with no non-calcified plaque. Multivariate analysis showed that the prevalence of DLP and NASH were independent risk factors for high-risk plaque (*p* < 0.05 each).

## 4. Discussion

This prospective single-center study showed that CAS and non-calcified plaque, a predictor of cardiovascular events, were more frequent in patients with NASH than in those with NAFL, which was diagnosed based on liver biopsy. In addition, NASH tended to be associated with CACS, whereas NAFL was not.

NAFLD may be a cardiovascular risk factor because a higher frequency of CVD was present in patients with NAFLD than in those without [20,21]. The association between NAFLD and atherosclerosis has been investigated in previous studies using the CAC score, which reflects the total coronary atherosclerotic burden. However, inconsistent results have been reported [22,23]. In addition, the absence of coronary calcification does not preclude the presence of significant CAD [12,24]. A recent study investigated the association between NAFLD and coronary atherosclerotic plaque using more comprehensive information regarding coronary atherosclerosis provided by CCTA [25]. However, the participants had not had their NAFLD confirmed by liver biopsy. Therefore, the present study evaluated the effect of the pathology associated with NAFLD on the risk of subclinical coronary atherosclerosis, not only on CACS but also on CAS and non-calcified plaque, as assessed using coronary artery CT.

Lipid disturbances, insulin resistance, and chronic inflammation are common features of the pathogenesis of NAFLD and CAD [26,27,28]. We previously reported that the reduction in the liver secretion of very low-density lipoprotein (VLDL) in patients with NASH is because of the reduction in VLDL synthesis secondary to the lower expression of microsomal triglyceride transfer protein [13,29,30]. The surplus triglycerides in these livers are therefore transported as triglyceride-rich lipoproteins (VLDL1), a precursor of sdLDL particles, which are more potent risk factors than LDL-C for atherosclerotic diseases, including CVD [31,32,33]. In addition, NAFLD and CVD progress because of the combination of insulin resistance, chronic inflammation, and oxidative stress, which are features of NASH rather than of NAFL. Therefore, we hypothesized that CAD is more common in patients with NASH than in those with NAFL, and the present study showed that patients with NASH had significantly worse coronary artery lesions than those with NAFL. This may be explained by higher sdLDL, insulin resistance, and chronic inflammation/oxidative stress in patients with NASH than in those with NAFL.

Coronary arteriosclerosis, evaluated using CACS, correlates with the concentration of a non-invasive fibrosis marker, the FIB-4 index, in patients with NAFLD. Our data show that CAD is worse in NAFLD patients with apparent fibrosis (fibrosis stage > 2). However, the multivariate analysis showed that the presence of NASH was a more significant risk factor for coronary artery lesions than fibrosis stage. Interestingly, a previous report showed that patients with NAFLD cirrhosis predominantly had liver-related events, whereas those with bridging fibrosis (fibrosis stage 3) predominantly had vascular events [34]. These findings suggest that NASH-related liver cirrhosis tends to impair liver function and reduce blood pressure, cholesterol concentration, and body mass, resulting in a lower incidence of vascular events than that observed in NASH patients with fibrosis stages 2–3.

This study had several limitations. First, the use of liver biopsy as the gold standard for assessing liver pathology was associated with sampling errors and intra- and interobserver variability, which are at least partly associated with the biopsy size. Second, the age and the percentage of gender were significantly different between NAFL and NASH groups in this cohort. Additionally, the patients included in the present study were referred from several hepatology centers in Japan that have a particular interest in NAFLD. For these reasons, some selection bias cannot be ruled out. Patient selection bias may also have occurred because of the result of liver biopsies being more likely to be performed for NAFLD patients at a risk of NASH. Finally, the study population could be different from the general population with NAFLD because the enrolled patients had chest pain and/or electrocardiographic abnormalities.

## 5. Conclusions

In this prospective study of patients with biopsy-diagnosed NAFLD undergoing CCTA, NASH was associated with a higher incidence of CAD than patient with NAFL, which implies a higher risk of cardiovascular events. However, these findings require corroboration in additional studies.

## Figures and Tables

**Figure 1 diagnostics-10-00129-f001:**
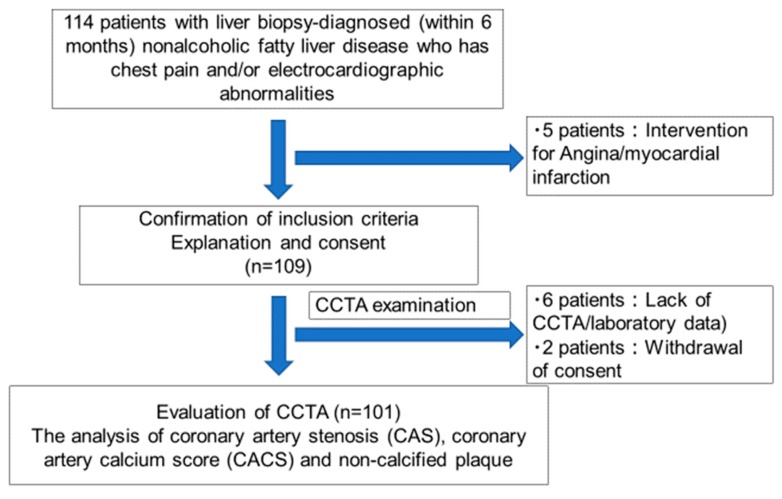
Study protocol.

**Figure 2 diagnostics-10-00129-f002:**
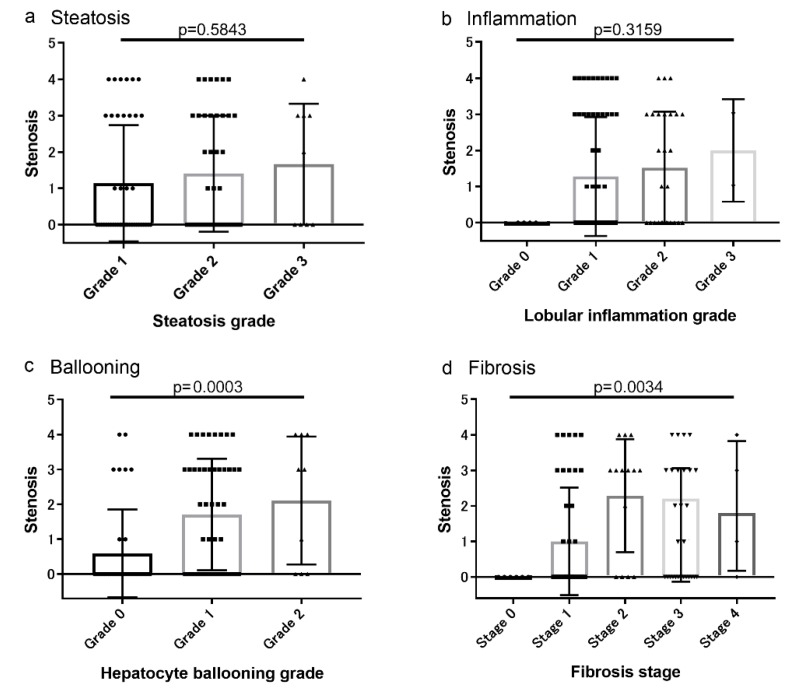
Association between the grade of coronary artery stenosis (CAS) and the histologic characteristics (**a**) Steatosis, (**b**) inflammation, (**c**) Ballooning, and (**d**) Fibrosis. The CAS grade was positively correlated with the hepatocyte ballooning grade and the stage of liver fibrosis.

**Figure 3 diagnostics-10-00129-f003:**
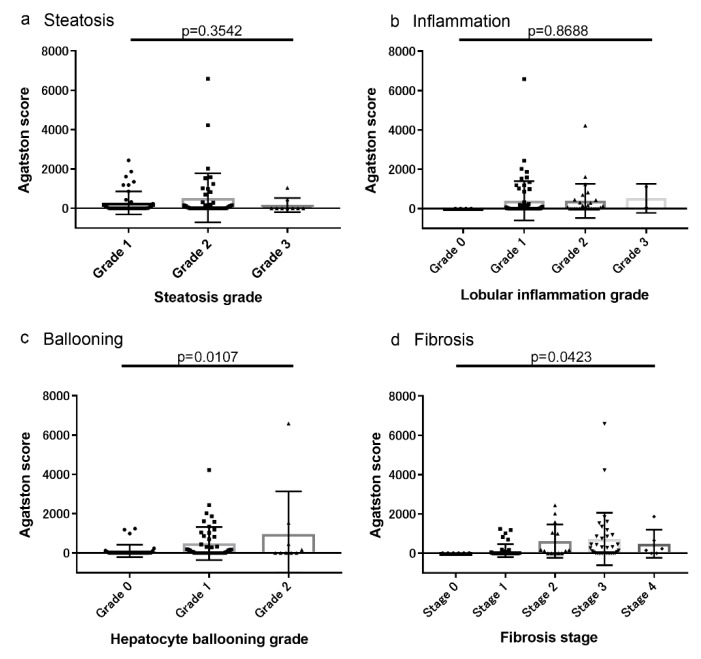
Association between the coronary artery calcium score (CACS) and the histologic characteristics (**a**) Steatosis, (**b**) inflammation, (**c**) Ballooning, and (**d**) Fibrosis. CACS was positively correlated with the hepatocyte ballooning grade and the stage of liver fibrosis.

**Figure 4 diagnostics-10-00129-f004:**
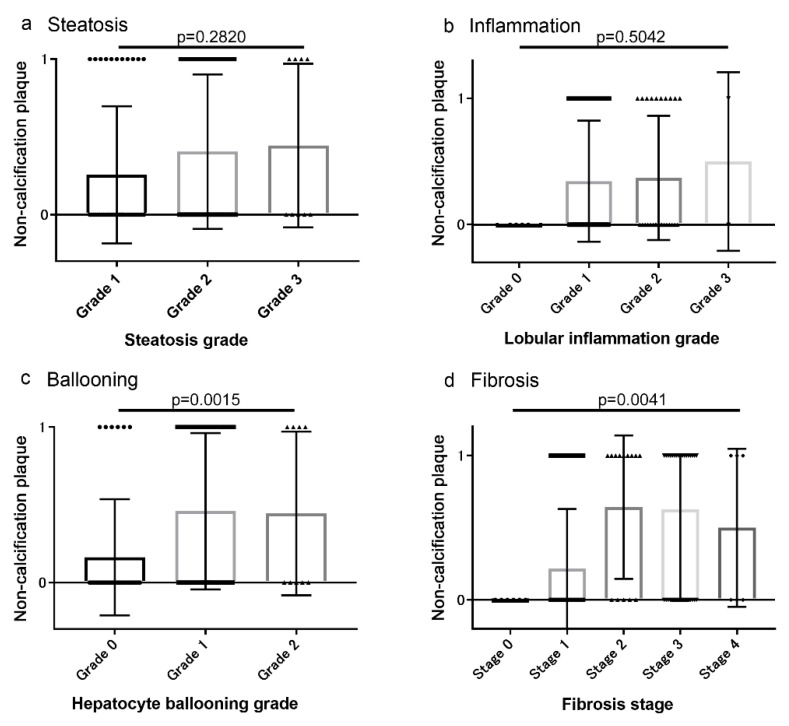
Association between non-calcified plaques and histologic characteristics (**a**) Steatosis, (**b**) inflammation, (**c**) Ballooning, and (**d**) Fibrosis. The presence of non-calcified plaques was positively correlated with the hepatocyte ballooning grade and the stage of liver fibrosis.

**Table 1 diagnostics-10-00129-t001:** Serological and histological characteristics in patients with NAFLD. *, statistical significance, *p* value < 0.05 in Univariate analysis.

	NAFL	NASH	*p* value *
Number (*n*)	41	60	
Age (years)	52.7 ± 13.9	59.6 ± 11.2	0.0090 *
Gender (male; female)	24; 17	26; 34	0.0471 *
Body mass index (kg/m^2^)	28.5 ± 4.2	27.8 ± 3.9	0.4803
AST (IU/l)	50.9 ± 33.1	51.1 ± 21.3	0.9749
ALT (IU/l)	73.6 ± 45.5	62.7 ± 30.5	0.1694
C-reactive protein (mg/l)	0.22 ± 0.35	0.19 ± 0.20	0.6555
Creatine (mg/dl)	0.73 ± 0.17	0.67 ± 0.17	0.1095
Fasting blood glucose (mg/dl)	117.6 ± 25.0	118.8 ± 30.0	0.8435
Fasting insulin (mU/l)	13.4 ± 7.09	18.6 ± 17.4	0.0915
HbA1c	6.23 ± 0.97	6.45 ± 1.08	0.3405
Diabetes mellitus	21 (51.2)	42 (68.3)	0.4714
Hypertension (%)	16 (39.0)	28 (46.7)	0.5188
Dyslipidemia (%)	24 (58.5)	46 (76.7)	0.3008
Steatosis grade			0.5734
5–33%	21	25	
33–66%	17	29	
>66%	3	6	
Lobular inflammation			<0.0001 *
None	6	0	
<2 foci per 200× field	32	32	
2–4 foci per 200× field	3	26	
>4 foci per 200× field	0	2	
Hepatocyte ballooning			<0.0001 *
None	38	0	
Few balloon cells	3	51	
Many balloon cells	0	9	
NAFLD activity score (NAS)	2.52 ± 0.86	4.22 ± 1.17	<0.0001 *
Fibrosis stage			0.0009 *
None	7	0	
Perisinusoidal or periportal	21	24	
Perisinusoidal and portal/periportal	3	13	
Bridging fibrosis	9	19	
Cirrhosis	1	4	

**Table 2 diagnostics-10-00129-t002:** Univariate analysis and multiple regression analysis of potential risk factors for coronary artery stenosis in patients with NAFLD. *, statistical significance, *p* value < 0.05 in Univariate and Multiple regression analysis, respectively.

Factors	Absence (n = 51)	Presence (n = 50)	Univariate *p* value * *p* < 0.05	Multiple *p* value * *p* < 0.05
Age (years)	52.4 ± 12.6	62.5 ± 10.8	0.0001 *	0.0049 *
Gender (male; female)	22; 29	29; 21	0.2201	
Body mass index (kg/m^2^)	28.0 ± 4.27	28.3 ± 3.73	0.8314	
Smoking score	120.8 ± 354.3	358.2 ± 454.6	0.0042 *	0.0147 *
Serum ALT levels	72.9 ± 44.9	62.8 ± 26.0	0.1730	
Serum GGT levels	90.3 ± 74.5	66.4 ± 41.5	0.0647	
Renal dysfunction (%)	3.9	9.3	0.2863	
Diabetes mellitus (%)	50.9	74.0	0.0354 *	0.6901
Hypertension (%)	33.3	51.1	0.0801 *	0.8215
Dyslipidemia (%)	56.9	79.0	0.0208 *	0.0373 *
Nonalcoholic steatohepatitis	43.1	82.0	0.0085 *	0.0102 *
Fibrosis stage	1.52 ± 1.06	2.34 ± 1.00	0.0002 *	0.0450 *

**Table 3 diagnostics-10-00129-t003:** Univariate analysis and multiple regression analysis of potential risk factors associated with Agatston score in patients with NAFLD. *, statistical significance, *p* value < 0.05 in Univariate and Multiple regression analysis, respectively.

Factors	Absence (score = 0) **(n = 50)	Presence (score > 0) **(n = 51)	Univariate *p* value *** *p* < 0.05	Multiple *p* value *** *p* < 0.05
Age (years)	51.6 ± 12.7	63.1 ± 9.6	<0.0001 *	0.0023 *
Gender (male; female)	23; 27	28; 23	0.3707	
Body mass index (kg/m^2^)	28.2 ± 4.19	28.1 ± 3.70	0.9470	
Smoking score	165.4 ± 377.2	309.8 ± 454.4	0.0857	
Serum ALT levels	73.1 ± 42.7	62.8 ± 29.8	0.1648	
Serum GGT levels	90.8 ± 74.9	66.4 ± 41.7	0.0588	
Renal dysfunction (%)	3.9	9.3	0.2863	
Diabetes mellitus (%)	44.0	80.4	0.0001 *	0.0232 *
Hypertension (%)	28.0	58.9	0.0016 *	0.2039
Dyslipidemia (%)	56.0	82.3	0.0037 *	0.0118 *
Nonalcoholic steatohepatitis	48.0	76.4	0.0029 *	0.0987
Fibrosis stage	1.62 ± 1.06	2.23 ± 1.07	0.0047 *	0.1892

**Table 4 diagnostics-10-00129-t004:** Univariate analysis and multiple regression analysis of potential risk factors associated with non-calcified coronary artery plaque in patients with NAFLD. *, statistical significance, *p* value < 0.05 in Univariate and Multiple regression analysis, respectively.

Factors	Absence **(n = 63)	Presence **(n = 38)	Univariate *p* value * *p* < 0.05	Multiple *p* value * *p* < 0.05
Age (years)	54.3 ± 12.8	62.2 ± 10.8	0.0012 *	0.0688
Gender (male; female)	28; 35	23; 15	0.1163	
Body mass index (kg/m^2^)	28.1 ± 3.91	28.3 ± 4.01	0.8136	
Smoking score	156.5 ± 384.2	373.9 ± 451.7	0.0114 *	0.0684
Serum ALT levels	68.8 ± 42.0	66.4 ± 26.9	0.7494	
Serum GGT levels	86.0 ± 69.3	66.5 ± 44.9	0.1532	
Renal dysfunction (%)	4.8	9.4	0.4058	
Diabetes mellitus (%)	55.6	73.7	0.0255 *	0.9736
Hypertension (%)	34.9	57.9	0.0240 *	0.6271
Dyslipidemia (%)	58.7	86.8	0.0020 *	0.0240 *
Nonalcoholic steatohepatitis	49.2	84.2	0.0003 *	0.0307 *
Fibrosis stage	1.67 ± 1.12	2.37 ± 0.94	0.0017 *	0.0947

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
