# Peer review of "Coronary Artery Disease is More Severe in Patients with Non-Alcoholic Steatohepatitis than Fatty Liver"

_diagnostics, 2020, doi:10.3390/diagnostics10030129_

Round 1
Reviewer 1 Report
Toshihiro Niikura et al. described that coronary artery lesions are more common in patients with NASH than in those with NAFL by using coronary computed tomography angiography (CCTA) and multi-slice CT. This interesting study provided further association between coronary artery disease and NAFLD, suggesting that patients with NASH should be closely monitored for coronary artery diseases. What I’m concerned about is why they particularly selected the patients with chest pain and/or electrocardiographic abnormalities (Figure 4). Additionally, the age and gender are significantly different between NAFL and NASH group (Table 1). These may introduce potential bias in this study and affect the reliability of these results.
Author Response
Thank you very much for your helpful and grateful suggestions. In accordance with these, we have revised our manuscript and feel that it is now greatly improved as a result.
Comments:
- What I’m concerned about is why they particularly selected the patients with chest pain and/or electrocardiographic abnormalities (Figure 4).
Response:
Thank you very much for your useful suggestions. Your comments are quite reasonable. Actually, we wanted to perform coronary computed tomography angiography (CCTA) in all patients undergoing liver biopsy. However, coronary computed tomography angiography (CCTA) is not usually performed in the patients without any heart symptoms such as chest pain or electrocardiographic abnormalities because of the use of radiation. Therefore, we enrolled only NAFLD patients with chest pain and/or ECG abnormalities. We described this point as a limitation because it may be related to bias (Page 7, line 162 to 163).
- Additionally, the age and gender are significantly different between NAFL and NASH group (Table 1). These may introduce potential bias in this study and affect the reliability of these results.
Response:
Thank you very much for your useful suggestions. Your comments are quite reasonable. We added these points as limitations in our revised manuscript (Page 7, line 158 to 161).
Reviewer 2 Report
The study showed that patients with biopsy-diagnosed NAFLD undergoing CCTA, NASH was associated with a higher incidence of CAD than in NAFL. No signigicant changes of the manuscript should be done.
Author Response
Thank you very much for reviewing our manuscript.